# A Non-Invasive Millimetre-Wave Radar Sensor for Automated Behavioural Tracking in Precision Farming—Application to Sheep Husbandry

**DOI:** 10.3390/s21238140

**Published:** 2021-12-06

**Authors:** Alexandre Dore, Cristian Pasquaretta, Dominique Henry, Edmond Ricard, Jean-François Bompa, Mathieu Bonneau, Alain Boissy, Dominique Hazard, Mathieu Lihoreau, Hervé Aubert

**Affiliations:** 1Laboratory for Analysis and Architecture of Systems, Toulouse University, CNRS, INPT, 31400 Toulouse, France; dhenry@laas.fr (D.H.); herve.aubert@toulouse-inp.fr (H.A.); 2Research Center on Animal Cognition (CRCA), Center for Integrative Biology (CBI), CNRS, University Paul Sabatier-Toulouse III, 31400 Toulouse, France; cristian.pasquaretta@univ-tlse3.fr (C.P.); mathieu.lihoreau@univ-tlse3.fr (M.L.); 3GenPhySE, Toulouse University, INRAE, ENVT, 31326 Castanet Tolosan, France; edmond.ricard@inrae.fr (E.R.); jean-francois.bompa@inrae.fr (J.-F.B.); dominique.hazard@inrae.fr (D.H.); 4URZ, INRAE, Petit-Bourg, 97170 Guadeloupe, France; mathieu.bonneau@inrae.fr; 5UMR Herbivores, Clermont University, INRAE, VetAgro Sup, 63122 Saint-Genès Champanelle, France; alain.boissy@inrae.fr

**Keywords:** radar sensors, radar signal processing, animal farming, computational ethology, signal classification, wavelet analysis

## Abstract

The automated quantification of the behaviour of freely moving animals is increasingly needed in applied ethology. State-of-the-art approaches often require tags to identify animals, high computational power for data collection and processing, and are sensitive to environmental conditions, which limits their large-scale utilization, for instance in genetic selection programs of animal breeding. Here we introduce a new automated tracking system based on millimetre-wave radars for real time robust and high precision monitoring of untagged animals. In contrast to conventional video tracking systems, radar tracking requires low processing power, is independent on light variations and has more accurate estimations of animal positions due to a lower misdetection rate. To validate our approach, we monitored the movements of 58 sheep in a standard indoor behavioural test used for assessing social motivation. We derived new estimators from the radar data that can be used to improve the behavioural phenotyping of the sheep. We then showed how radars can be used for movement tracking at larger spatial scales, in the field, by adjusting operating frequency and radiated electromagnetic power. Millimetre-wave radars thus hold considerable promises precision farming through high-throughput recording of the behaviour of untagged animals in different types of environments.

## 1. Introduction

Behavioural research increasingly requires automated recording and analyses of animal movements [1]. This is exemplified by emerging methods for high-throughput monitoring and statistical analyses of movements that enable the quantitative characterisation of behaviour on large numbers of individuals, the discovery of new behaviours, but also the objective comparison of behavioural data across studies and species [2,3]. These quantitative approaches are particularly powerful to study inter-individual behavioural variability or personalities in animal populations [4]. In livestock, for instance, large-scale genetic selection programmes are based on the measurements of several hundreds (if not thousands) of farm animals [5]. Many behavioural tests have been developed to assess personality traits in these animals [6], with some applications in breeding programmes, for instance to discard the more aggressive individuals [7]. However, in these studies behavioural measures are frequently obtained from direct observations by the experimenters or farmers [8], which considerably limits the possibility to quantify behavioural traits at the experimental or commercial farm level.

Animal tracking methods involving on-board devices, such as Global Positioning Systems (GPS) [9], radio telemetry [10], radio frequency identification (RFID) [11] or harmonic radar [12], are hardly suitable for detailed high throughput behavioural phenotyping due to the limited accuracy and duration of measurements. Best available approaches therefore involve image-based analyses [13]. So far however, these techniques often require large computational resources to fit the classification model and to process images [14], and are sensitive to light variation [15]. Moreover, video processing using machine learning is typically limited to the detection of one type of target (e.g., the focal animal species), which means that other potentially important information in the signal (e.g., the presence of a farmer) is ignored.

Recently, Frequency-Modulated Continuous-Wave (FMCW) radars operating in the millimetre-wave frequency band have been proposed for the automated tracking of the behaviour of a large diversity of animals (sow: [16], bees: [17]; sheep: [18]). In this approach, it was shown it is possible to record one-dimensional movements (distance to radar) of individual sheep in an arena test [18]. Tracking animals with FMCW radars has the great advantage of being non-invasive (does not require a tag), insensitive to light intensity variations, and fast (does not require large memory resource). FMCW radars therefore provide considerable advantages for the development of automated high-throughput analyses of behaviour in comparison to more conventional approaches like video and infrared cells. The radar signal processing does not require fitting a model to detect targets, which relaxes the need to collect thousands of data before application. In addition, it offers the possibility to detect targets placed behind a non-transparent wall, which can be used to hide the tracking device, or to study the effect of physical obstacles on an animal’s behaviour.

Here we report a millimetre-wave FMCW radar system for the automated tracking and analysis of the 2D trajectories of freely moving animals. We illustrate our approach with the analysis of the movements of 58 sheep in an experimental farm. The measurements were performed during a behavioural test commonly used to estimate the sociability of individual sheep in genetic selection [8,19]. First, we compared the estimate of the sheep position with the radar and standard video tracking and infrared cells. Second, using the radar data we identified new behavioural estimators that could be used for large-scale behavioural phenotyping. Third we showed that the radar system can also operate for long-distance tracking, in the field, by adjusting radar emission frequency and radiated electromagnetic power.

## 2. Material and Methods

### 2.1. Sheep

We ran the experiments in July 2019 at the experimental farm la Fage of the French National Research Institute for Agriculture, Food, and Environment (INRAE), France (43.918304, 3.094309). We tested 58 lambs (29 males, 29 females) *Ovis aries* with known weight (range: 12–31.3 kg) and age (range: 59–88 days). Ewes and their lambs were reared outdoor on rangelands. After weaning, lambs were reared together outside and tested for behaviour 10 days later. This delay enabled the development of social preferences for conspecifics instead of preference for mother.

All the lambs were previously tested in a “corridor test” to estimate their docility towards humans. Briefly, the test pen consisted of a closed, wide rectangular circuit (4.5 × 7.5 m) with opaque walls [8]. A non-familiar human entered the testing pen and walked at constant speed through the corridor until two complete tours had been achieved. The corridor was divided into 6 virtual areas. Every 5 s, the areas in which the human and the animal were located were recorded and the mean distance separating the human and the lamb was calculated. The walking human also recorded with a stopwatch the total duration when he could see the head of the lamb to discriminate between fleeing and following lambs. The reactivity criteria towards an approaching human was constructed by combining both distance and duration measurements (for more details see [20]). The higher the resulting variable (i.e., “docility” variable in the present study), the more docile the animal.

### 2.2. Arena Test

We measured sheep behaviour in a standard protocol (“the arena test”) used to assess the sociability of sheep through measures of inter-individual variability in social motivation in the absence or presence of a shepherd [8,19]. A sheep (focal sheep) was introduced in the pen (2 m × 7 m) (Figure 1A) (for more details see [21]). Three other sheep from the same cohort (social stimuli) were placed behind a grid barrier, on the opposite side of the arena entrance. The test involved three phases (Figure 1B):-In phase 1, the focal sheep could explore the arena for 15 s and see its conspecifics through a grid barrier;-In phase 2, visual contact between the focal sheep and the social stimuli was disrupted using an opaque panel pulled down from the outside of the pen for 60 s. This phase was used to assess the sociability of the sheep towards its conspecifics;-In phase 3, visual contact between the focal sheep and its conspecifics was re-established and a human was standing still in front of grid barrier for 60 s. This phase was used to assess the sociability of the focal sheep towards conspecifics in presence of a immobile human.

Sets of 2 infrared cells were placed at the height of the sheep’s body and every meter along the arena test to define 7 virtual areas of 1 m. Analyses of the data resulting from the activation of the infrared cells by the sheep were performed with Fortran algorithms to compute longitudinal displacements of the sheep in the device.

### 2.3. Data Collection

We measured the displacement of the focal sheep in phases 2 and 3 of the arena test (phase 1 is the initiation phase) using three automated tracking systems: (1) infrared sensors, (2) a video camera, and (3) a millimetre-wave FMCW radar. During the measurement, an experimenter also recorded the number of high-pitched bleats by the focal sheep, a proxy of sociability [8]. A proximity score was computed as the time spent in each virtual area weighted according to the virtual area delimited by the infrared receptor in such a way that a high score indicated high proximity to conspecifics [20]. Crossing rate measured the number of virtual areas crossed during arena test phases 2 and 3.

### 2.4. Video and Radar Tracking

We compared the efficiency of the radar system and standard video tracking for monitoring the 2D movements of the sheep. For the video tracking, we placed a camera on one end of the arena (opposite to entrance side, Figure 1B). The camera was elevated 2 m above ground in order to film the entire arena, producing black and white images of size 720 p × 576 pixels every 25 ms. Sheep movements were tracked in 2D. For image processing, we applied a detection algorithm using the state-of-the-art image object detector tiny-YOLO V3 (You Only Look Once) network, which is a version of the YOLO model adapted for faster processing allowing 244 images of 0.17 mega pixels (416 × 416 pixels) per second (on a TITAN X graphics card) [23]. This Convolutional Neural Network (CNN) was pre-trained on the PASCAL Visual Object Classes Challenge dataset [24]. YOLO detected all the objects on the image, including the focal sheep, possibly some parts of the background and the human when entering inside the arena. To differentiate between the sheep and non-sheep detected objects, we used another CNN, Alexnet, that we parameterized using transfer learning [25]. A set of 40 sheep and 40 non-sheep images were used to re-train the network. Finally, for some images the focal sheep was not detected, especially when it was located at the opposite of the camera. In these cases, the location of the sheep was extrapolated by continuing the trajectories with a constant speed between the two known locations.

For the radar tracking, we placed a millimetre-wave FMCW radar (Figure 1C, see technical characteristics in Table 1) at one end of the arena test (i.e., entrance side, Figure 1B). The radar was setup outside of the test pen behind a Styrofoam wall transparent to millimetre-waves [26]. The transmitting antenna array radiated a repetition over time of a so-called chirp (i.e., a saw-tooth frequency-modulated signal [27]). The chirp was backscattered by the targeted focal sheep, but also by the surrounding scene which provides undesirable radar echoes called the electromagnetic clutter. The total backscattered signal was then collected by the receiving antenna array and processed to mitigate the clutter and to derive the sheep 2D trajectory from radar data. In the millimetre-wave frequency range, the detectability of the sheep depends mainly on the bandwidth of the frequency modulation, the beamwidth of the radar antennas, and the radiated electromagnetic power [27].

Processing of radar data included two main steps. First, we extracted the position of the animal. Next, we computed behavioural parameters to characterize the movement of the animal. We extracted the distance of the focal sheep to the radar and its direction in the horizontal plane of the scene. To mitigate the electromagnetic clutter, we estimated the mean value and standard deviation of the radar signal in absence of the sheep and we derived the signal, denoted by D, from the signal S delivered by the radar in presence of the animal, as follows:$$D\left(t,r,\theta \right)=\frac{S\left(t,r,\theta \right)-mean\left(r,\theta \right)}{std\left(r,\theta \right)}$$
where r is the radar-to-sheep separation distance, mean is the time-averaged radar signal at the range r and angular position θ, std is the time-standard deviation of the radar signal. Figure 1D shows an example of position estimations of a sheep over time after removing the electromagnetic clutter.

### 2.5. Extraction of New Behavioural Parameters Form the Radar Data

We used the radar estimated 2D trajectories to extract new behavioural parameters characterizing sheep movements using three approaches.

1: Behavioural classes;

We statistically identified broad classes of behaviour using Gaussian Mixture Models (GMM). First, we divided the trajectories into time windows of 1 s for each sheep and for each experimental phase. Next, we extracted movement parameters from each window: average speed, sinuosity (total displacement over distance between the first position and the last) and total displacement distance. Then, because the social stimuli (i.e., the three conspecifics) were located at one end of the corridor, we split the speed vector into two components: along the two lateral walls of the corridor and across the two longitudinal walls. Finally, to derive behavioural classes we performed a GMM on the extracted movement parameters for each lamb [28]. The number of classes (i.e., the number of Gaussians to be used) was determined by comparing models using 1 to 15 classes. We selected the model with the lowest Akaike score, which represents the model with the features best explaining the parameter under consideration [29]. The GMM was performed using the Python package scikit-learn [30]. We estimated the rate of time spent in each movement classes for the two phases.

2: Behavioural transitions;

We determined behavioural changes over time using Ricker wavelet processing [32]. Wavelet processing consists in filtering the sheep position signal using a wavelet as a filter [33]. The use of wavelet analysis to describe animal behaviour was previously used in [34]. This type of filtering is applied to several time scales, thus allowing the detection of a change in the direction and speed of the sheep, depending on when the changes occur or the duration of the change. Our aim was to determine the precise moments when the focal sheep changed its way of moving, which was estimated using the spectrum described by each scale of the used wavelet. We observed that the number of local maxima in the wavelet transform coefficients is sensitive to the number of changes in the way of moving and the size of the wavelet will determine if the change is global or punctual. This estimation of changes was done on the lateral and longitudinal movement and for the two last phases of the experiment.

3: Space coverage;

We investigated the space occupied across time by the focal sheep using heatmaps representing the areas the sheep spent time in during the measurements. The use of heatmaps to describe animal behaviour was previously used in [35]. We partitioned the arena into a grid of 80 virtual zones of 44 × 40 cm^2^ each (i.e., 16 partitions along the arena length and 5 partitions along the arena width). We chose this grid dimension because it is the width of a small lamb [36]. We counted the number of zones (i.e., the heatmap score) the focal sheep remained in for more than 200 ms. This count was used to extract behavioural features for the two last phases of the experiment.

### 2.6. Outdoor Radar Tracking

We ran outdoor experiments in order to demonstrate the applicability of our radar system for the tracking of sheep in field conditions. These measurements were done in an open space with no obstacles (60 m × 15 m asphalt place). A human experimenter moved within the radar catching area in order to induce animal movements. We tested one female sheep. To enhance detection range to 40 m, we used a FMCW radar with the lower operating frequency of 24 GHz. At fixed transmitted power, lower frequencies enable reduction of the free-space attenuation of the radiated electromagnetic power [27]. The gain due to the free-space attenuation is 10.13 dB.

### 2.7. Statistical Analyses

We ran all analyses using the programming environment R [37]. Raw trajectory data extracted from radar and video measures are available in Appendix A.

Analysis of new movement features

We tested the influence of sheep characteristics (docility, and sociability) in interaction with the two test phases on the proportion of time spent in the behavioural classes using a generalized Linear Mixed Model with binomial family error distribution. We tested all possible dual interactions of each variable with the test phase. Three-way Interactions were excluded to avoid over-fitting of the model [38]. Sheep identity was included as a random effect. We ran a model selection on all feature combinations (docility, sociability the phases and their interactions) using the Akaike score. The model with the lowest score was retained as the best model. When the second best model have an AIC score equivalent to the best model (i.e., when the difference is lower than 2) an average model was performed with those that have equivalent AIC. We used a similar procedure to test the influence of the sheep individual characteristics on continuous wavelet transforms estimated on lateral and longitudinal movements (Gaussian family error distributions) and heatmaps (Poisson family error distribution).

Classification of behavioural types;

To improve the interpretation of the sheep behaviour in the corridor, we reduced our four movement features (proportion of fast movements, changes in longitudinal and transversal movements, space coverage) for phases 2 and 3, using a Principal Component Analysis (PCA). The PCA was performed using the R package FactoMineR [39]. We explored afterward whether our new automated estimators could be used to replace estimators recorded manually using a General Linear Model (GLM, using the R package stats) approach.

## 3. Results

### 3.1. Radar Tracking Is Faster and More Accurate Than Video Tracking

To test the efficiency of the radar tracking system, we compared the data obtained from the infrared cells, the video and the radar. This efficiency was estimated by comparing the proximity score estimated using the infrared cell, video and radar detection but also by using the crossing rate estimated by the infrared cell and the mean speed along the longitudinal axis estimated by the video and radar detection. We analysed data from 58 individuals (29 males, 29 females). Both data collected by the radar and the video enabled to capture information given by infrared cells with high fidelity. Proximity scores and crossing rates obtained from infrared cells were positively correlated with data obtained from the radar (Pearson correlation; proximity: r = 0.77, *p* < 0.001; crossing rate: r = 0.87, *p* < 0.001) and the video (Pearson correlation test; proximity: r = 0.91, *p* < 0.001; crossing rate: r = 0.34, *p* < 0.001).

Radar tracking had additional advantages over video tracking in terms of data processing (Table 2). The radar produced two times more measures per second. Radar processing was also much faster (50 frames per second for radar and 4 for video processing) and therefore, it may be used for real time analyses. Radar measurement data were of similar size as video measures (ROM), but required approximately seven times less memory (RAM) to process. Finally, radar processing did not require a learning phase with important data collection and a time-consuming training phase that can last several hours just for the adaptation of the model, or several days if the network is not trained beforehand.

### 3.2. New Behavioural Indicators from the Radar Data

The following analyses were made on the 58 sheep. The 2D radar trajectory data offered the opportunity for high resolution analyses of sheep movements.

Behavioural classes: detection of slow and fast movements

In order to classify the different types of movements exhibited by the sheep, we applied the GMM procedure to statistically identify behavioural classes from the trajectory data. We found four behavioural classes (Figure 2A):

Class 1 (51.3% of the measures) was characterized by null or slow movements (“slow movements”);

Class 2 (35.48% of the measures) was characterized by fast movements with low sinuosity (“fast movement”);

Class 3 (10.2% of the measures) was characterized by fast movements with high sinuosity (“fast tortuous”);

Class 4 (3.01% of the measures) was characterized by slow movements with high sinuosity (“slow tortuous”).

Each of the two behavioural classes with strong sinuosity (classes 3 and 4) represented less than 10% of all data. We thus focused our analyses on slow and fast movements only (classes 1 and 2). We tested the effects of the individual characteristics of sheep on the rate of time spent in each in the two main behavioural classes using GLMMs. The best (using Akaike criterion) model (See Appendix A) retained the docility, sociability indicators and the phase of the test to explain the two main behavioural classes extracted by the radar, i.e. the rate of slow movement and fast movement. In phase 3, all the sheep tended to move less than in phase 2 (estimate = −1.24, std. = 0.008, *p* < 0.001). In phase 2, highly sociable sheep moved less than little sociable sheep (estimate = −0.11, std. = 0.015, *p* < 0.001). This trend was reduced in phase 3 for both sociable and docile sheep (sociability: estimate = −0.12, std. = 0.039, *p* < 0.001 docility: estimate = 0.16, std. = 0.0074, *p* < 0.001) (Appendix A and Figure 2).

Wavelet analysis: detection of erratic behavioural transitions;

Our second approach to describe the sheep behaviour was to quantified changes in movements (i.e., variation in speed, direction, or both) through time. This was done using continuous wavelet analyses (Figure 3). We tested the effects of the individual characteristics of sheep on the frequency of these changes using GLMMs and model selection (Appendix A). When considering longitudinal displacements (i.e., wavelet Y) along the arena device (Table 4), we found that highly sociable sheep made more changes in the pattern of displacement during both phases of test (estimate = 16.98, std. = 4.68 *p* < 0.001) (Figure 3A,C). In general the movements were less erratic in phase 2 than in phase 3 (estimate = −91.50, std. = 9.07, *p* < 0.001). When considering transversal movements (i.e., wavelet X) across the arena device (Table 4), we found that sheep made more changes in the way of displacement during phase 2 than phase 3 of test (estimate = −53.15, std. = 8.26, *p* < 0.001) (Figure 3B,D). However, this trend was reduced for the docile sheep (estimate = 19.19, std. = 7.11, *p* = 0.009).

Heatmap analyses: Detection of spatial coverage

Finally we quantified the spatial coverage by individual sheep (number of zones occupied in the arena) using heatmaps (Figure 4). Overall, the sheep used 2.37 (std. 1.03) time less space in phase 3 than in phase 2. We tested the effects of the individual characteristics on the number of zones in which the sheep spent more than 200 ms using GLMMs and model selection. Here we describe the most explanatory model considering AIC, but the three best models gave a similar trend on the sheep behaviour (see Appendix A), so that an average model was ultimately performed using the models with n difference of AIC lower than 2 with the best model. Using a spatial resolution of the grid similar to the dimension of a lamb body size (i.e., dimension: 0.44 × 0.40 m; example Figure 4A) revealed that sheep tended to use less space in phase 3 than in phase 2 (estimate = −0.765, std. = 0.053, *p* < 0.001), and that highly sociable sheep used more space in phase 2 than less sociable sheep (estimate = 0.048, std. = 0.024, *p* = 0.043). It also showed that most docile sheep used less space in phase 2 than less docile sheep (estimate = −0.066, std. = 0.031, *p* = 0.0389) but the phenomenon was reduced in phase 3 (estimate = 0.099, std. = 0.046, *p* = 0.032) (Table 5). Therefore, the influence of sociability on spatial coverage decreased in phase 3.

### 3.3. Sheep Behavioural Phenotype

We explored whether the new movement features extracted from the radar data could capture information from behavioural traits measured manually by the experimenter in the arena test. We focused on docility and sociability. We ran a PCA based on the eight behavioural measures extracted from the radar data in phase 2 and phase 3: proportion of fast movements (class 1) out of all movements (class 1 + class 2), longitudinal movements (wavelets Y), transversal movements (wavelets X) and space coverage (heatmaps). We retained two PCs using the Kaiser–Guttman criterion [40]. PC1 explained 30.65% of the variance and PC2 explained 19.31% of the variance (Table 6). The eigenvalues associated to the 3 first components are: 2.8928914, 1.7375911, 0.9738257. PC1 was positively associated with all behavioural variables (Figure 5A). Sheep with high PC1 values moved more often fast, made more changes in the way of displacement, and used more zones than sheep with low PC1 values. We therefore interpreted PC1 as a “movement” component. PC2 was positively associated with the four behavioural variables of phase 3 and negatively associated with the four behavioural variables of phase 2 (Figure 5A). Sheep with high PC2 values showed a more important increase of time spent moving fast, of the frequency of changes in the way of displacement, and numbers of zones occupied between phase 2 and phase 3 than sheep with low PC2 levels. We interpreted PC2 as a variable of “movement in response to social isolation”. Using PC1 and PC2, we investigated contribution of the docility and sociability of the sheep on these components. It showed that the first was linked to the sociability (estimate = 0.2690, std. = 0.1054, *p* = 0.0135) and the second was linked to docility (estimate = 0.28296, std. = 0.1111, *p* = 0.0137). The link between PC1 and docility and PC2 and sociability was not significant.

### 3.4. Outdoor Radar Tracking

To demonstrate that our radar tracking system could be used at larger spatial scales, in the field, we sat up a radar with a lower operating frequency in an outside corridor (10 × 60 m; Figure 6A). We successfully monitored the 2D trajectory of one sheep over a maximum distance of 45 m the backscattering signal was not detectable using one radar measurement (Figure 6B). The presence of a human to induce sheep movement did not deteriorate sheep tracking (Figure 6C).

## 4. Discussion

Research in animal behaviour increasingly requires automated monitoring and annotation of animal movements for comparative quantitative analyses [2,3]. Here we introduced a radar tracking system suitable to study the 2D movements of sheep indoor and outdoor, within a range of 45 m. A summary of the method is shown on Figure 7. The system is non-sensitive to light variations, compatible with real time data analyses, transportable, fast processing and adaptable to various species and experimental contexts. Moreover, it does not require tags or transponders to track animals. It is therefore suitable for the collection of large sets of behavioural data in an automated way required in many areas of biological and ecological research, as well as applied ethology for precision farming as illustrated here.

We recently used FMCW radars to track the behaviour of sheep [18], pigs [16] and bees [17]. Here, however, for the first time, we demonstrate the applicability of this approach to monitor 2D trajectories of untagged walking animals within a range of 45 m. Others methods can be used to estimate the sheep position, such as video detection [24] which can detect sheep in 2D up to 20 m but with a precision from 50 cm (at 5 m) to 1 m (at 20 m) and GPS detection [41], but this requires to equip the animals with transponders. We showed that the radar acquisition system has several advantages over these more conventional methods, and in particular video tracking. It collects more data per second (50 measures per second for the radar versus 25 for the video), requires less RAM (524 Kb for one radar measurement versus 3.7 Mb for one video frame). It also requires 10 times less processing time (e.g., does not require to train neural networks) and generates less false detection rates (15% of false detection for video processing and 5.2% for radar processing). Importantly, the radar is not dependent on brightness and can be used for outside tracking over long distances by adjusting operating frequencies. It also enables the tracking of individualized animals without tags, based on the size and shape of the radar echoes of the different targets.

Our application of radar-based tracking to behavioural phenotyping of sheep shows that the radar analysis is consistent with current semi-automated analyses (i.e., infrared sensors and video). Using the radar, we found that sheep tend to have a greater displacement in phase 2 than in phase 3 of the arena test. This agrees with previous studies showing that sheep are more active when socially isolated from conspecifics [20,21]. Higher behavioural activity in a social isolation context, for instance through locomotion and vocalization behaviours, may be interpreted as the way for the isolated animal for searching for social contact with conspecifics as described in the ewe-lamb relationships [42] or between familiar lambs [43].

In addition, the high resolution 2D, in theory 5 cm in range and 6° in azimuth, trajectories obtained from the radar enabled identification of new behavioural estimators that could greatly benefit the fast and automated identification of behavioural phenotypes. For example, our application of unsupervised behavioural annotation to identify statistically significant behaviours by sheep in the arena test showed that sheep exhibit less fast movements in phase 3 than in phase 2. The wavelet analysis, considering the way that the sheep moves (i.e., referred to here as “way of displacement”) revealed the occurrence of “erratic” displacements. Here low erratic displacements corresponded to displacements showing a constant speed whereas high erratic displacements corresponded to a high level of alternation in slow and fast displacements. These erratic displacements may be linked to the sociability and/or docility of sheep. Finally, space occupation analysis showed that individuals exploit narrower areas in phase 3 than in phase 2 of the arena test. All these results are consistent with previous observations using semi-automated recording methods. Indeed, social isolation from conspecifics (i.e., phase of test 2) resulted in the expression of on average higher behavioural activity (i.e., individual variability exists), including displacements, than in presence of conspecifics and a motionless human (i.e., phase of test 3). The higher displacement activity during social isolation resulted in a higher exploration of the arena whereas, in presence of conspecifics and a motionless human, lambs showed limited displacement. The combination of these new automatically computed estimators appears to be complementary to behavioural traits of interest that were until now measured (i.e., for instance no or slight relationship with sociability or docility) and could be used for more detailed characterization of animal behavioural profiles. Note, however, that this first study is based on relatively low sample sizes (58 individuals) and further measurements are needed to verify the biological trends observed on a much larger number of sheep.

Beyond the case study of the arena test described here, our system could be tuned to suit a large diversity of animal sizes and experimental contexts. Several ways can be considered. For instance, the range and resolution of detection could be improved using different radars. Here, we had to place the radar at 1 m from the arena fences in order to illuminate and monitor the entire arena. Antennas with larger beamwidth may allow placing the radar on the arena fences. Moreover, the detection was limited to a few meters, but it is possible to detect a sheep at tens of meters using a radar operating at a lower frequency (24 GHz) and/or transmitting higher electromagnetic power. It is also possible to improve radar detection by using more antennas. Indeed, by multiplying the number of antennas, we multiply the number of signal estimations and then the noise from the radar can be decreased. The same radar technology could be used to track individuals in groups over longer distances in open fields, for instance to explore the mechanisms underpinning social network structures and collective behaviour [44]. The processing of the radar signal can also be improved for tracking large number of sheep simultaneously by using deep radar processing but this would require the use of a large amount of annotated data to train the neural networks [45]. Individual tracking within groups could also be improved with non-invasive passive tags that depolarize radar signal in specific directions [46]. Note that at the moment, we do not know the long-term effects of the use of millimetre waves on these animals and this should be investigated in further studies.

## 5. Conclusions

We demonstrated the feasibility of tracking a sheep in a restricted area using a millimetre-wave FMCW radar. This detection is possible even if each wall of the arena backscatters the transmitted electromagnetic signal. This radar tracking system can also be advantageously used to extract features that are correlated to the movement of the sheep and can estimate if it is erratic, fast and the space occupied in the corridor. In contrast to other short-range tracking methods, our radar detection approach does not require pre-annotated data and can be applied in real time. This flexibility holds considerable premises for tracking the behaviour of animals of various sizes and environments in a wide range of contexts and research fields.

## Figures and Tables

**Figure 1 sensors-21-08140-f001:**
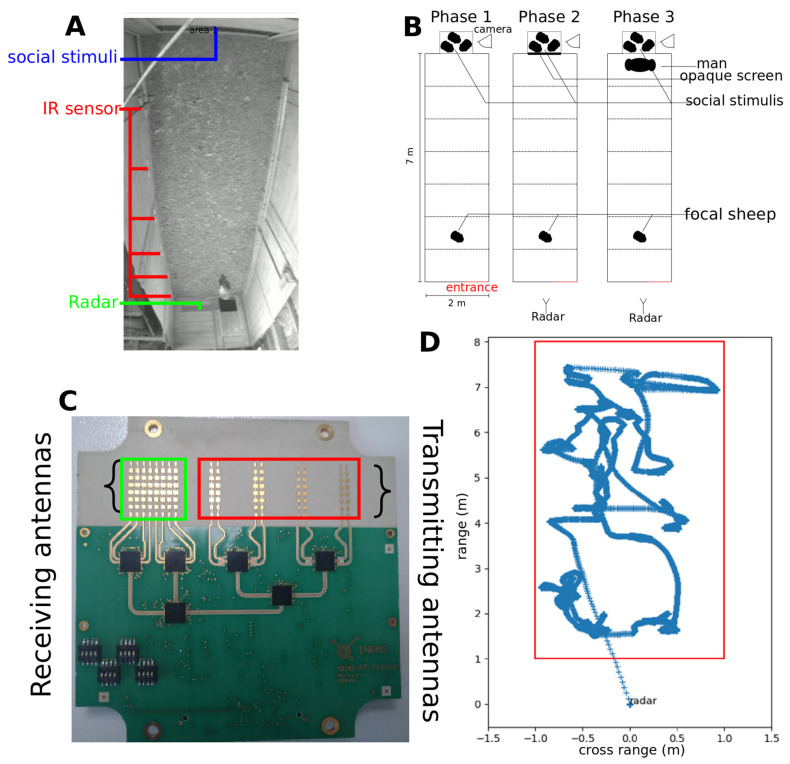
Corridor test. (**A**) Top view of the focal sheep and the social stimuli in the corridor (example image extracted from video data). (**B**) Schematic representation of experimental phases 1, 2 and 3. (**C**) Image of the FMCW radar frontend (phot credit AD). Each rectangle corresponds to patch [22]. (**D**) Example of a trajectory of a sheep obtained with radar tracking after removing the clutter and normalizing the estimated value. The red rectangle represent the pen walls.

**Figure 2 sensors-21-08140-f002:**
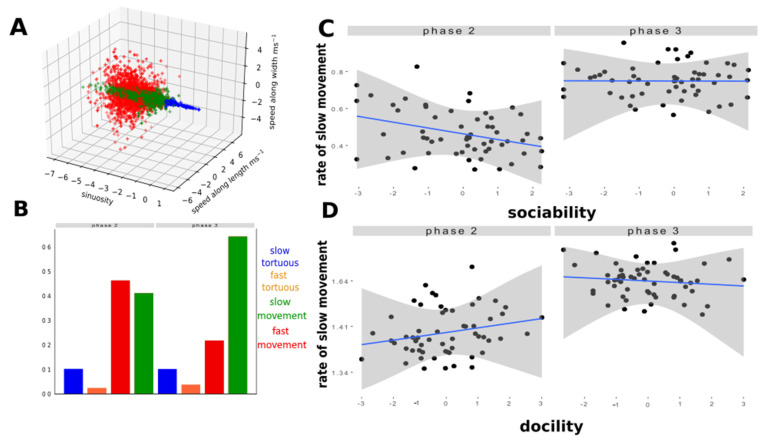
Analyses of behavioural classes. (**A**) Distribution of the four behavioural classes after a Gaussian Mixture Model. (**B**) Frequency of behavioural classes during phase 2 and phase 3 of the corridor test. (**C**) Correlation between the proportion of time spent in slow movements and the sociability score of sheep during phase 2 and 3 (see details of models in Table 3). (**D**) Correlation between the proportion of time spent in slow movements and the docility score of sheep during phase 2 and 3. N = 58 sheep.

**Figure 3 sensors-21-08140-f003:**
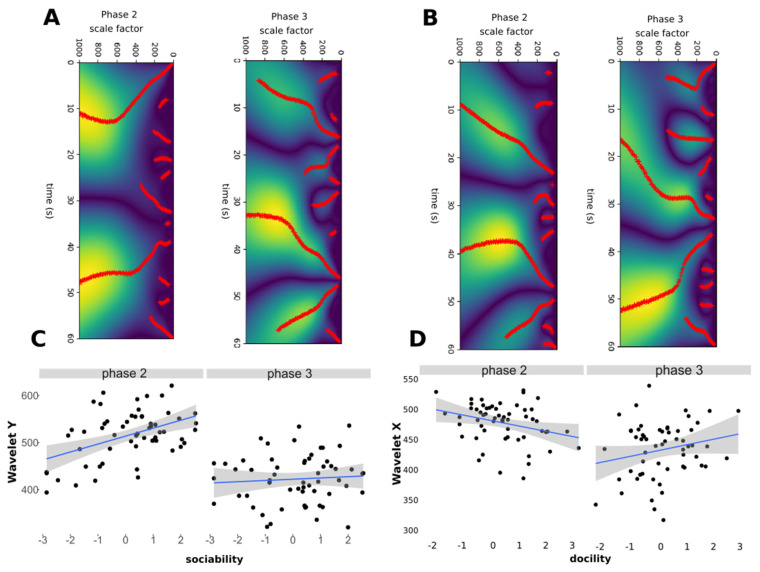
Wavelet analyses. (**A**) Example of wavelet transform for lateral movements (X). Red dots correspond to the detection of a change in the displacement at scale factor and time position (i.e., a local maxima of the wavelet transform of the signal position). (**B**) Example of wavelet transform for longitudinal movements (Y). (**C**) Relationship between the number of local maxima (red dots in (**A**,**B**)) in the wavelet extraction and the degree of sociability of sheep during phases 2 and 3. (**D**) Relationship between the number of wavelets and the degree of docility of sheep during phases 2 and 3. See details of models in Table 4. N = 58 sheep.

**Figure 4 sensors-21-08140-f004:**
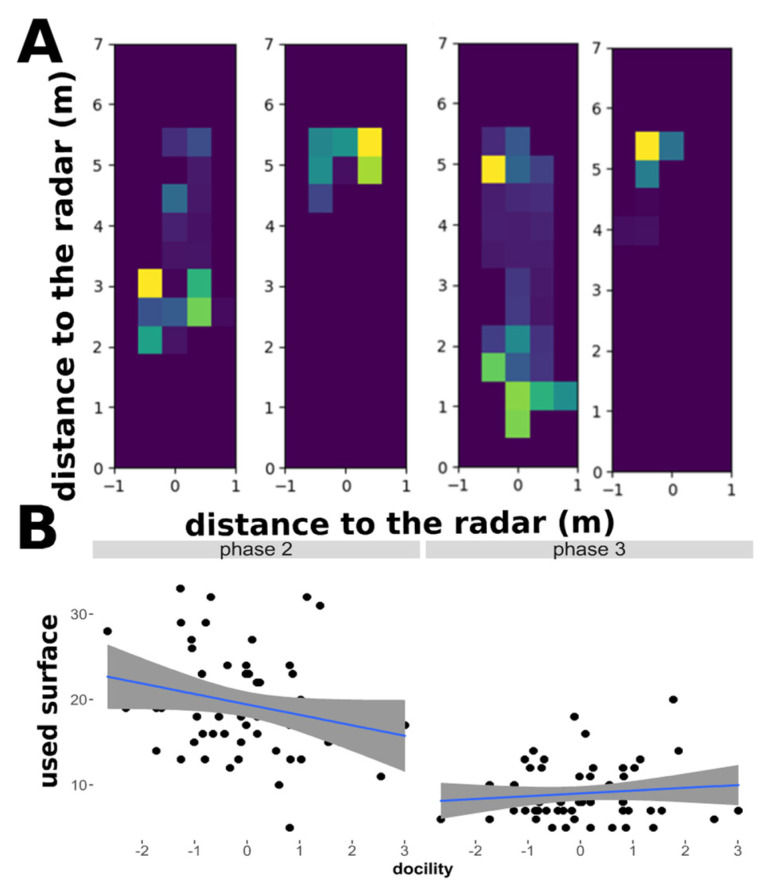
Heatmap analyses. Relationship between the numbers of areas occupied by the lambs and the degree of docility in phase 2 and phase 3. (**A**) Resolution grid (cell dimension: 0.44 × 0.40 m). (**B**) Relationship between the surface used by the sheep the degree of docility of sheep during phases 2 and 3. See details of models in Table 5. N = 58 sheep.

**Figure 5 sensors-21-08140-f005:**
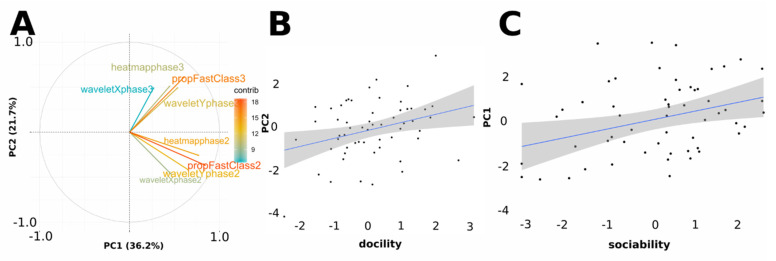
(**A**) Correlations between the two first components (PCs) of the principal component analysis (PCA). Arrows represent the eight behavioural variables on PC1 (movement speed) and PC2 (movement increase between phases). Contribution of variables to the variance explained is color-coded. Each data point represents the PC1 and PC2 scores of a given lamb (N = 58). (**B**) Relationship between PC1 and sociability. (**C**) Relationship between PC2 and docility. Blue lines represent linear models (see main text). N = 58 sheep.

**Figure 6 sensors-21-08140-f006:**
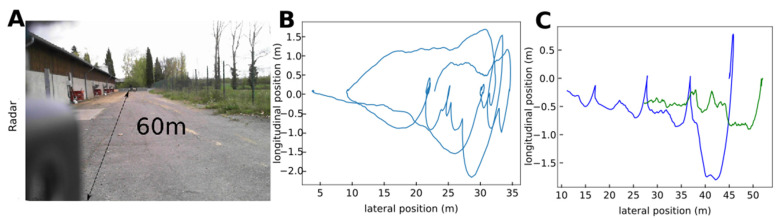
(**A**) Picture of the outside corridor used for radar tracking of a sheep (credit AD). The radar was positioned 60 m from the end of the corridor. (**B**) Example of trajectory of a sheep derived from the radar data. (**C**) Example of trajectory of a sheep (red) and a man (green) derived from the radar data.

**Figure 7 sensors-21-08140-f007:**
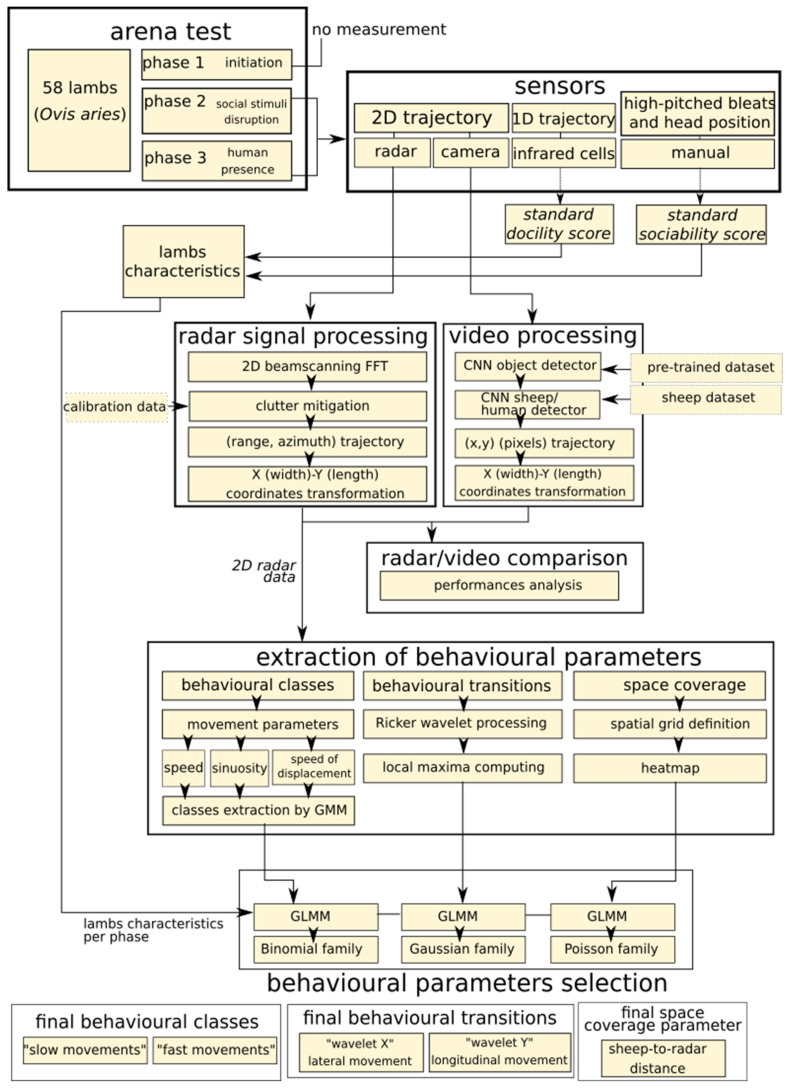
Summary of the method described in the study, from the behavioural test and the acquisition of the data with radar to the extraction of the new behavioural parameters form the trajectory data.

**Table 1 sensors-21-08140-t001:** Technical characteristics of the FMCW radar used for indoor tracking [22] and outdoor tracking [31].

Name	Indoor Tracking	Outdoor Tracking	Note
Operating frequency	77 GHz	24 GHz	This frequency is also called the carrier frequency of the frequency-modulated signal transmitted by the radar
Modulation Bandwidth	3 GHz	800 MHz	Frequency interval, centred at the operating frequency, used for the saw-tooth frequency modulation of the transmitted signal
Ramp time	256 µs	1 ms	Up-ramp duration of the saw-tooth frequency-modulated signal (or chirp duration)
Repetition time	50 ms	30 ms	Period of the transmitted frequency-modulated signal (or chirp repetition interval)
Number of linear arrays of the transmitting antenna array	4	1	One linear array composed of 8 × 2 rectangular patches radiating elements
Number of linear arrays of the receiving antenna array	8	2	Eight linear arrays composed of 8 rectangular patches radiating elements
Main lobe beamwidth of the transmitting antenna array in the horizontal plane	50°	58°	Angular range (or field of view) of the radar illumination in the horizontal plane
Transmitted power	100 mW	100 mW	Power delivered at the input terminals of the transmitting array antenna (the radiated power is defined as the product of the transmitted power by the efficiency of the antenna)

**Table 2 sensors-21-08140-t002:** Comparison of data processing characteristics with radar and video tracking systems.

Tracking Method	Radar	Video
Number of measures per second	50	25
Read Only Memory (ROM) for all measures of a sheep	151 Mo	62 Mo
Random Access Memory (RAM) per measure	524 Kb	3.7 Mb
Processing time per measure	<20 ms	250 ms
Distance to target centre	1.1 m	1.5 m

**Table 3 sensors-21-08140-t003:** Analyses of behavioural classes. Results of the best GLMM (binomial family, after model selection—see Appendix A). The model tested the effects of phase, docility, sociability, and dual interaction of each variable with phase, on the proportion of time spent in fast movements (behavioural class 2). Lamb identity was included as a random factor. Significant effects (*p* < 0.05).

	Estimate	Std. Error	z Value	Pr (>|z|)
(Intercept)	0.11	0.055	2.08	0.037
Sociability	0.13	0.039	3.47	<0.001
phase 3	−1.24	0.0086	−144.04	<0.001
Docility	−0.11	0.047	−2.43	0.015
sociability:phase 3	−0.12	0.0061	−19.90	<0.001
Docility: phase 3	0.16	0.0074	21.31	<0.001

**Table 4 sensors-21-08140-t004:** Wavelet analyses. Results of the best GLMM (Gaussian family, after model selection—see details in Appendix A). The model tested the effects of phase, docility, sociability, and binary interactions of each variable with phase, on the number of wavelets. Lamb identity was included as a random factor. Significant effects (*p* < 0.05) are shown in bold. Wavelet Y: longitudinal movements. Wavelet X: transversal movements.

Wavelet Y	Estimate	Std. Error	Df	t Value	Pr (>|t|)
**(Intercept)**	**514**	**6.64**	**110**	**77.3**	**<0.001**
**sociability**	**17**	**4.68**	**110**	**3.63**	**<0.001**
**phase 3**	**−91.5**	**9.07**	**55**	**−10.1**	**<0.001**
docility	−3.12	5.72	110	−0.545	0.587
Sociability:phase 3	−14.4	6.4	55	−2.25	0.05
Docility:phase 3	4.7	7.81	55	0.602	0.55
**Wavelet X**	**Estimate**	**Std. Error**	**df**	**t Value**	**Pr (>|t|)**
**(Intercept)**	**467**	**6.04**	**110**	**77.3**	**<0.001**
sociability	0.526	4.26	110	0.124	0.902
**phase 3**	**−53.2**	**8.26**	**55**	**−6.43**	**<0.001**
Docility	−9.61	5.2	110	−1.85	0.0673
Sociability:phase 3	7.36	5.82	55	1.26	0.212
**Docility: phase 3**	**19.2**	**7.11**	**55**	**2.7**	**<0.05**

**Table 5 sensors-21-08140-t005:** Heatmap analyses. Results of the best GLMM (Gaussian family, after model selection—see details in Appendix A). The model tested the effects of phase, docility, sociability, and dual interactions of each variable with phase, on the number of areas where the lamb spent more than 1 s. Lamb identity was included as a random factor. Significant effects (*p* < 0.05) are shown in bold.

Heatmap	Estimate	Std. Error	z Value	Pr (>|z|)
**(Intercept)**	**2.95**	**0.037**	**79.00**	**<2 × 10^−16^**
**docility**	**−0.066**	**0.031**	**2.07**	**0.039**
**phase 3**	**−0.77**	**0.053**	**14.27**	**<2 × 10^−16^**
**sociability**	**0.048**	**0.023**	**2.022**	**0.043**
**phase 3: docility**	**0.099**	**0.046**	**2.15**	**0.032**
phase 3: sociability	−0.020	0.038	0.52	0.60

**Table 6 sensors-21-08140-t006:** Eigenvalue for each component (PC) of the Principal Component Analysis using the eight behavioural features extracted using the radar tracking.

Component	Eigenvalue	Variance Explained
PC 1	2.893	30.65
PC 2	1.738	19.31
PC 3	0.974	13.04
PC 4	0.833	9.27
PC 5	0.564	7.20
PC 6	0.492	6.69

## Data Availability

The dataset used to compare radar features and behavioural scores are available in Appendix A.

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
