# Peer review of "A Non-Invasive Millimetre-Wave Radar Sensor for Automated Behavioural Tracking in Precision Farming—Application to Sheep Husbandry"

_sensors, 2021, doi:10.3390/s21238140_

Round 1

Reviewer 1 Report

This paper deals with a basically new application for MMW/THz sensing. It is novel, and certainly worthy of publication. I think it can be improved further by clarifying the following:

Lines 118 and 213 - which thermal imaging spectral bands were used for heatmaps? Is one more advantageous than others?

Different frequencies are used for indoor and outdoor tracking. Lower frequencies can reduce free space attenuation [lines 226-7]. Why? By how much typically? I suspect the difference is rather small.

Lines 203 and 205 - what do the authors mean by "way of moving"? Do they mean direction of motion?

Section 3 - advantages of radar tracking are noticable. I suggest mentioning them, as well as some additional important results, in the abstract. It should help spark interest. 

Author Response

Dear reviewer,

Thank you for your comment on the manuscript, please find below the response to the different notes

All the best,

Alexandre Dore

This paper deals with a basically new application for MMW/THz sensing. It is novel, and certainly worthy of publication. I think it can be improved further by clarifying the following:

  1. Lines 118 and 213 - which thermal imaging spectral bands were used for heatmaps? Is one more advantageous than others?

> We are not sure what it means. We did not use thermal imaging in our study. The heatmaps were computed from the radar measurements by estimating the time spent by each sheep in each part of the arena. Text L217 now reads: “We investigated the space occupied across time by the focal sheep using heatmaps representing the areas the sheep spent time in during the measurements. The use of heatmaps to describe animal behavior was previously used in [35]. We partitioned the arena into a grid of 80 virtual zones of 44x40cm² each (i.e. 16 partitions along the arena length and 5 partitions along the arena width). We chose this grid dimension because it is the width of a small lamb [36]. We counted the number of zones (i.e. the heatmap score) the focal sheep remained in for more than 200ms. This count was used to extract behavioral features for the two last phases of the experiment. »

  1. Different frequencies are used for indoor and outdoor tracking. Lower frequencies can reduce free space attenuation [lines 226-7]. Why? By how much typically? I suspect the difference is rather small.

> We used another frequency based on preliminary results showing that the maximal range with a 77GHz radar was around 15 meters and the maximal range with a 24GHz radar is around 45 meters. The difference in free space attenuation is about 10dB. This is explained L233.

  1. Lines 203 and 205 - what do the authors mean by "way of moving"? Do they mean direction of motion?

> Text L330 now reads « Our third approach to describe the sheep trajectory was to quantified changes in the way of moving (variation in speed, direction, or both) through time. »

  1. Section 3 - advantages of radar tracking are noticable. I suggest mentioning them, as well as some additional important results, in the abstract. It should help spark interest.

> Done, Text L21 now reads “In contrast to more conventional video tracking systems, radar tracking requires low processing power, is independent on light variations and has more accurate estimations of animal positions due to a lower misdetection rate.”

Reviewer 2 Report

In this paper by Alexandre Dore et al, entitled “A Non-invasive Millimeter-wave Radar Sensor for Automated Behavioral Tracking in Precision Farming – Application to Sheep Husbandry”, the authors have introduced introduce a new automated tracking system based on millimeter-wave radars for real time robust and high precision monitoring of untagged animals, which requires low processing power and is independent on light variations. The novelty of the paper is high. And the present submission is good organized and written.

In my opinion, the paper should be accepted in present form.

Author Response

Dear reviewer,

Thank you for your comment on the manuscript.

All the best,

Alexandre Dore

Reviewer 3 Report

A new automated tracking system based on millimeter wave radars for real-time robust and high precision monitoring of untagged animals is presented in the research. The proposed system requires low processing power and independent on light variation. The system is validated through various experiments. The article is well written, however there are some suggestions which should be implemented to improve the quality of the publication.

  1. The organization of the article should be presented at the end of the introduction section.
  2. The sections 2 and 3 contain sub-sections. At the starting of the section these sub-sections should be described in few lines to create the reader’s interest.
  3. The authors have presented several advantages of the proposed method. What are the drawbacks of exposure of such a high frequency on the animals’ health and the humans present to take care of the animals. 
  4. Please use separate heading for conclusion.

Author Response

Dear reviewer,

Thank you for your comment on the manuscript, please find below the response to the different notes

All the best,

Alexandre Dore

A new automated tracking system based on millimeter wave radars for real-time robust and high precision monitoring of untagged animals is presented in the research. The proposed system requires low processing power and independent on light variation. The system is validated through various experiments. The article is well written, however there are some suggestions which should be implemented to improve the quality of the publication.

The organization of the article should be presented at the end of the introduction section.

> Done, see L72.

The sections 2 and 3 contain sub-sections. At the starting of the section these sub-sections should be described in few lines to create the reader’s interest.

> Done, see L139. L290 L329

The authors have presented several advantages of the proposed method. What are the drawbacks of exposure of such a high frequency on the animals’ health and the humans present to take care of the animals. 

> Text L505 now reads « Note that at the moment, we donnot know the long-term effects of the use of millimeter waves on these animals and this should be investigated in further studies ».

Please use separate heading for conclusion.

> Done (L508).